# Hydrothermal Effects of Freeze-Thaw in the Taklimakan Desert

**Liu Xinchun [1], Kang Yongde [2], Chen Hongna [3] and Lu Hui [4],***

[1] Key Laboratory of Tree-Ring Physical and Chemical Research, China Meteorological Administration, Urumqi 830002, China; liuxch@idm.cn
[2] State Key Laboratory of Eco-Hydraulics in Northwest Arid Region of China, School of Water Resources and Hydroelectric Engineering, Xi'an University of Technology, Xi'an 710048, China; kyd0115@yeah.net
[3] Xinjiang Environmental Monitoring Station, Urumqi Environmental Monitoring Center, Urumqi 830001, China; liuxinchun2001@163.com
[4] Key Laboratory of Ecology of Rare and Endangered Species and Environmental Protection, College of Environment and Resources, Guangxi Normal University, Guilin 541004, China
* Correspondence: luhui1008@163.com

**Abstract:** The Taklimakan Desert, also known as the "Sea of Death", is the largest desert in China and also the world's second largest remote desert. The road crossing the Taklimakan Desert is the longest desert road in the world and has been the center of the Silk Road since ancient times. Based on field observation data (November 2013 to May 2014) collected from the Tazhong and Xiaotang stations, we studied the interannual and diurnal variations of soil temperature, soil moisture content, and surface heat fluxes during different freezing and thawing periods. The annual and daily changes of soil temperature, soil moisture content, and surface energy fluxes at different freezing and thawing stages were analyzed. We illustrated the coupling relationship between water and heat in freezing-thawing soil in the Taklimakan Desert. We established a coupling model of soil water and heat during freezing and thawing. During the soil freezing period, the soil temperatures at different depths generally trended downward. The temperature difference between the Tazhong station and the Xiaotang station was 4~8.5 °C. The freezing time of soil at 20 cm depth occurred about 11 days after that at 10 cm depth. The effect of ambient temperature on soil temperature gradually weakened with the increase of soil depth. With the occurrence of the soil freezing process, the initial soil moisture contents at 5 cm, 10 cm, 20 cm, and 40 cm depths at the Xiaotang station were 6%, 10%, 29%, and 59%, respectively, and those at the Tazhong station were 5%, 3.6%, 4.4%, and 5.8%, respectively. As the ambient temperature decreased, the freezing front continued to move downward and the liquid soil water content at each depth decreased. The desert highway is closely related to the economic development and prosperity of southern Xinjiang. Therefore, it is important to maintain and inspect the safety and applicability of freeze-thaw zones and avoid casualties from vehicles and personnel.

**Keywords:** seasonally frozen soil; soil moisture; soil temperature; hydrothermal condition

## 1. Introduction

Climate change driven by greenhouse gas emissions has attracted the attention of governments and scientists. Among many climate change issues, research on soil freezing-thawing and the hydro-thermal cycle has become popular in recent years [1].

Frozen soil accounts for about 75% of the land area in China, which also includes most of the buildings and roadbeds. The total area of seasonally frozen soil is roughly 5.14 million square kilometers, accounting for approximately 53% of the total land area of China. It is widely distributed above 24° N, especially in northwestern China. The freezing and thawing process can alter the soil structure by the phase change of soil pore space shrinkage-expansion induced by water, which destroys soil aggregates [2,3]. In addition to human factors, surface temperature affects freezing and thawing, which damages the soil structure [4–6]. The freeze-thaw (F-T) cycle is complicated [7]. During the F-T period, soil structure is damaged and shows frost heave and settlement. After that, the F-T cycle will

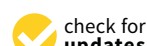



eventually affect the bearing capacity of the foundation. Therefore, the shear capacity of soil will destroy the buildings connected to it. Surface temperature also affects atmospheric circulation. It is the medium linking the atmosphere and the earth's surface.

At the early stage of soil thawing, ice and water in the soil frequently transform from one form to another. In this stage, the latent heat of the surface is absorbed and released. Oscillation of heat inside the soil and the corresponding F-T cycles cause changes in energy and water exchanges between the soil and the atmosphere [8,9]. Surface albedo, evapotranspiration, soil infiltration, sensible and latent heat fluxes, radiation fluxes, and exchanges of energy and momentum between the subsurface and the atmosphere are also affected by changes in runoff and vegetation conditions during soil F-T cycles. The soil freezing and thawing process has various impacts on climate change [10]. Soil moisture can affect many ecological processes, e.g., soil erosion, plant growth, and vegetation restoration [11,12]. Spatial variation of soil temperature has been studied at different sampling scales in different regions. Therefore, the quantitative analysis of spatial variation of soil temperature and its scale dependence is becoming an important research topic in soil and water science.

The main characteristics of soil temperature, evaporation, and freezing and thawing in the Taklamakan Desert region need to be investigated. The Taklimakan Desert Highway crosses the world's second largest nomadic desert and is also the largest mobile desert in China. Its construction effectively reduces the transportation cost between cities in Xinjiang and improves the timeliness of transportation.

Previous studies have demonstrated that water migration accompanying the freezing and thawing process of the subgrade surface can lead to frost heave and thaw settlement diseases. In other words, the moisture content of the soil at different depths will also affect the water and heat conditions of the subgrade. The surface soil of the desert is the interface for heat exchange with the atmosphere, and its water and heat exchanges are mainly controlled by climatic conditions, solar radiation, and soil characteristics.

Under the action of vehicle load and overburden load, a series of issues (e.g., road surface cracking, subsidence, tilting, and subgrade subsidence) can occur. Due to the lack of field observations, the application of empirical models is greatly limited. Hence, it is difficult to implement these models in the study of F-T patterns in the Taklimakan Desert. Existing studies mostly focused on the physical and chemical properties of soil under freezing and thawing conditions and the freezing and thawing pattern of permafrost regions in the Qinghai-Tibet Plateau. However, there are few studies on the freezing and thawing process of the Taklimakan Desert. The study on the characteristics and mechanism of soil hydrothermal coupling in the F-T process is essential for revealing the hydrological process of the Taklimakan Desert and establishing the soil hydrothermal coupling model.

The systematic experimental base of the atmospheric environment observation field in the Taklimakan Desert of the China Meteorological Administration is in the center of the Taklimakan Desert. It is the only atmospheric environment observation and experimental station in the world that is located more than 200 km deep in the hinterland of the remote desert. It lies deep in the Asian continent and far away from the sea. It is characterized by an inland warm temperate arid desert climate, arid climate, sparse vegetation, complex dunes, and solid sand movement. The research objective of this paper is to utilize the existing observations to clarify the F-T pattern of soil in desert areas and understand how it affects the surface energy. The observation data were collected from Tazhong and Xiaotang stations from November 2013 to May 2014. The temporal and spatial variation characteristics of temperature, moisture content, and surface energy at different soil depths during freezing and thawing period were analyzed.

## 2. Materials and Methods

### 2.1. Study Area

The Taklimakan Desert is the world's largest remote desert and has an area of about 33 square kilometers [13]. There is a 519 km desert highway connecting the north and south

of Xinjiang. With the influence of the Qinghai-Tibet Plateau, the climate of the Tarim Basin is seasonally dry. It has a typical temperate continental climate characterized by extremely low annual precipitation (below 50 mm) from the eastern to the central desert [14]. The desert is in the south of the Kunlun mountains and north of the Tianshan mountains. The study area is located in the hinterland of the desert at Tazhong and Xiaotang (Figure 1), and the area is mainly covered by fine and very fine sands.

Field experiments were conducted from November 2013 to May 2014. To obtain the soil moisture content and temperature information, measuring instruments were installed at 5 cm, 10 cm, 20 cm, and 40 cm below the ground surface (0 cm). The data of soil temperature and moisture content at these depths were recorded using a CR1000 data recorder.

Soil sensible and latent heat fluxes were measured using a hot plate (HPF01, Campbell Scientific company, Logan, UT Wohlwend Engineering), while radiative fluxes (short-wave and long-wave) were measured using a four-component net radiometer (CNR) mounted at 2 m above the ground.

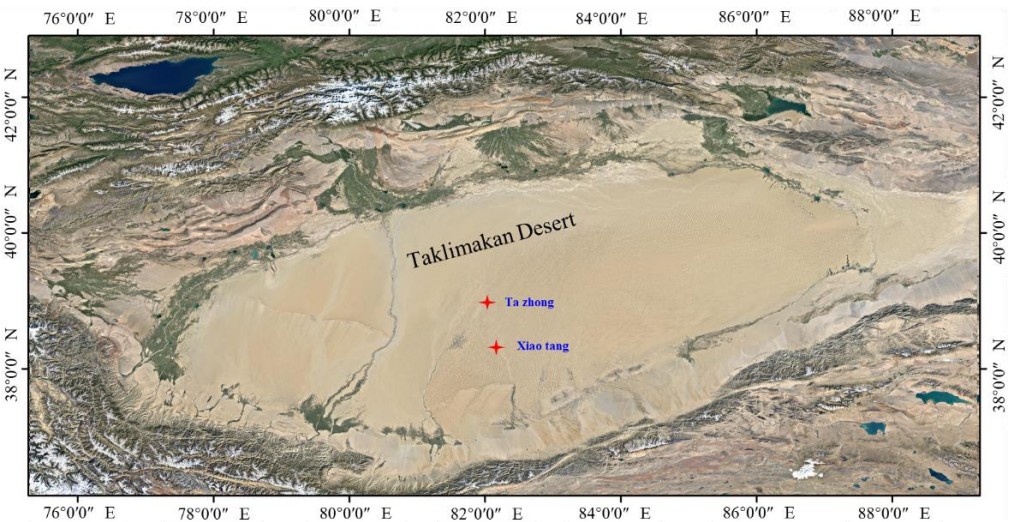

**Figure 1.** Map of the study area.

### 2.2. Data

In the field experiment, field data (including soil temperature, humidity, heat and radiative fluxes) were measured in order to clarify the patterns of soil freezing and thawing in the study area.

### 2.2.1. Soil Temperature and Moisture

Daily means of soil temperature and moisture content based on 24-hour means of field data can be calculated using:

$$T_d = \frac{\sum_{i=0}^{23} \mathrm{h}_i}{24} \tag{1}$$

$$\mathrm{M}_d = \frac{\sum_{i=0}^{23} \mathrm{m}_i}{24} \tag{2}$$

where $T_d$ is the average daily soil temperature, $\mathrm{M}_d$ is the average daily soil moisture, $\mathrm{h}_i$ is the measured soil temperature at time, $i$, in a day, and $\mathrm{m}_i$ is the measured soil moisture at time, $i$, in a day.

### 2.2.2. Sensible and Latent Heat Fluxes

The sensible heat flux (*H*) and the latent heat flux (LE) can be obtained from the 30-min mean covariance:

$$H = \rho C_p \overline{w'T'} \tag{3}$$

$$LE = L_v \rho \lambda \overline{w'q'} \tag{4}$$

where $\rho$ is the atmospheric density (kg.m$^{-3}$), $C_p$ is the specific heat capacity of the atmosphere at the constant pressure (J/kg.K), $w'$ is the velocity of the vertical wind (m.s$^{-1}$), $T$ is the air temperature (K), Lv is the latent heat of vaporization (w.m$^{-2}$), and q is the specific humidity of the air. (kg/kg).

### 2.2.3. Ground Heat Flux R$_n$

The heat flow plate can be installed at a certain depth. In general, the ground heat flux can be directly measured. With that, a one-dimensional soil heat conduction equation can be expressed as:

$$R_n = R(Z) + C_V \int_0^Z \frac{\partial T(z)}{\partial t} dz \tag{5}$$

In Equation (5), the first term on the right-hand side of the equation represents the soil heat flux at different depths, and the second term on the right-hand side represents the soil heat flux between the surface and the depth under time advancement. At 5 cm and 7.5 cm depths, Equation (5) can be written as Equations (6) and (7):

$$R_n = R(5 \text{ cm}) + C_V \times (0.025 \times \frac{\partial T_{0 \text{ cm}}}{\partial t} + 0.025 \times \frac{\partial T_{2.5 \text{ cm}}}{\partial t}) \tag{6}$$

$$R_n = R(7.5 \text{ cm}) + C_V \times (0.025 \times \frac{\partial T_{2.5 \text{ cm}}}{\partial t} + 0.05 \times \frac{\partial T_{5 \text{ cm}}}{\partial t}) \tag{7}$$

## 3. Results

### 3.1. Soil Temperature as Affected by F-T

Figures 2 and 3 show the changing patterns of soil temperature at different times during the freezing and thawing period. The soil temperatures at 5 cm, 10 cm, 20 cm, and 40 cm depths were measured during field experiments at Tazhong and Xiaotang stations in November 2013 and May 2014, respectively. During freezing and thawing, soil temperature changes with the ambient temperature. The direct characteristic of soil is a F-T phenomenon. During the freezing/thawing period, the ambient environment can significantly influence the shallow and middle soils (5 cm, 10 cm, 20 cm), and eventually affect the temperatures of the shallow and middle soils. Deep soil (40 cm) also varies with the ambient environment. Soil changes during the freezing process; soil temperatures at different depths overall showed a downward trend. From November to December, 2013, the temperatures at 5–40 cm depths at XiaoTang station were lower than those at Tazhong station. The temperature difference near the surface of the two stations was about 4~8.5 °C.

The diurnal variation of soil temperature is similar at the same depth. At 20 cm and 40 cm depths, the two sites had the largest daily variation. At the Xiaotang station, the freezing times of soils at 5 cm and 10 cm depths were very close on 23 November, 2013. The freezing time of the soil at 20 cm depth occurred relatively late on 8 December, 2013 (i.e., 11 days after the freezing time at 10 cm depth). The soil at 40 cm depth entered the freezing period in December 2013. Consistent with a study conducted by [10], some energy is consumed when soil freezes, resulting in small changes in soil temperature. Whilst the freezing date at the Tazhong station was brought forward, especially at the depths of 5 cm and 10 cm, the temperature variation trends of these two soil layers were consistent with the temperature variation of the ambient environment. The temperature of the ambient environment had some effect on the soil temperature. However, such effect weakened with the increase of soil depth. The lag phenomenon occurred in the middle and deep soil temperatures with the change of temperature of the ambient environment.

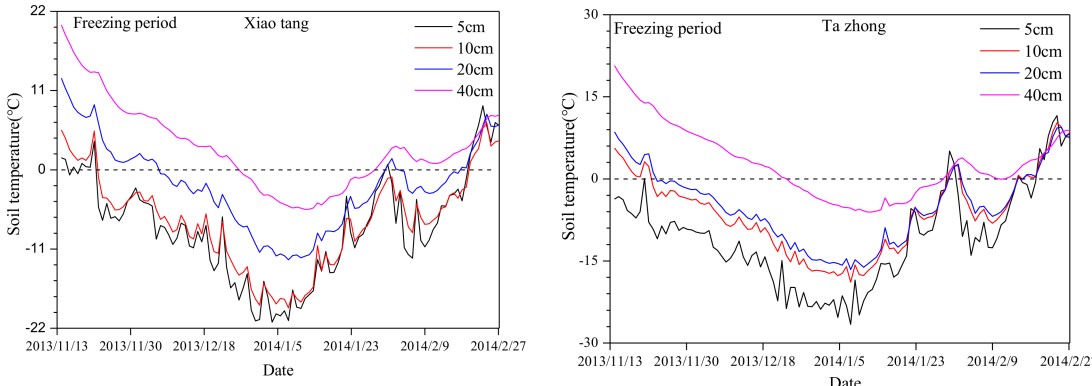

**Figure 2.** Temperature change at the Tazhong station and the Xiaotang station during the freezing periods.

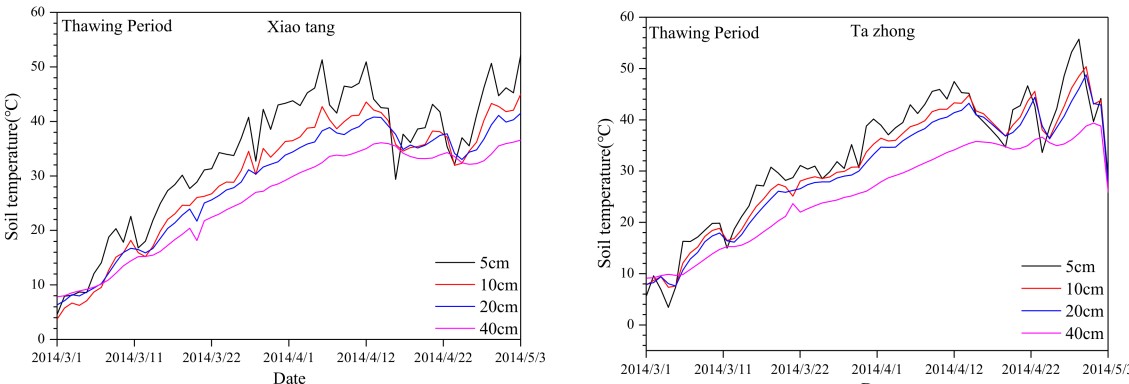

**Figure 3.** Soil temperature changes during thawing periods.

During the process of soil thawing, the soil temperatures at different depths generally trended upward. In early March, soil temperatures at 5 cm, 10 cm, 20 cm, and 40 cm depths at Xiaotang station were 4.7 °C, 3.8 °C, 6.4 °C, and 7.8 °C, respectively, while soil temperatures at similar depths were 5.7 °C, 7.8 °C, 7.9 °C, and 9.1 °C, respectively. The soil temperature at each soil depth increased with the gradual increase of the ambient temperature. In early April, soil temperatures at 5 cm, 10 cm, 20 cm, and 40 cm depths rose to 51.3 °C, 42.7 °C, 38.9 °C, and 33.6 °C. the difference in temperature change of the soil at different soil depths is 38.6, 38.9, 32.5, and 25.8 °C. The difference in the tower is taken as 47.5, 38.3, 35.7, and 26.9 °C. It can be seen that the temperature of each layer of soil consistently interacted with the trend of ambient temperature. The influence of ambient temperature on the soil temperature gradually weakened with the depth of the soil layer.

### 3.2. Soil Moisture as Affected by F-T

Transformation of water and ice always occurs during seasonal soil freezing and thawing. When the heat in the soil is released to the ambient environment in the form of heat transfer and convection and the ambient temperature reaches the freezing temperature of the soil, the soil moisture will begin to freeze. Conversely, when the soil temperature reaches the thawing temperature of the ice, the solid water will gradually transform into liquid water. Moisture content of the soil during the freezing and thawing periods at 5 cm, 10 cm, 20 cm, and 40 cm soil depth is shown in Figure 4. During the freezing and thawing periods, the shallow soils (5 cm, 10 cm) are significantly affected by the ambient environment, so the moisture of the shallow soil fluctuates with time, and the medium and deep soils (20 cm, 40 cm) have relatively small changes in water content.

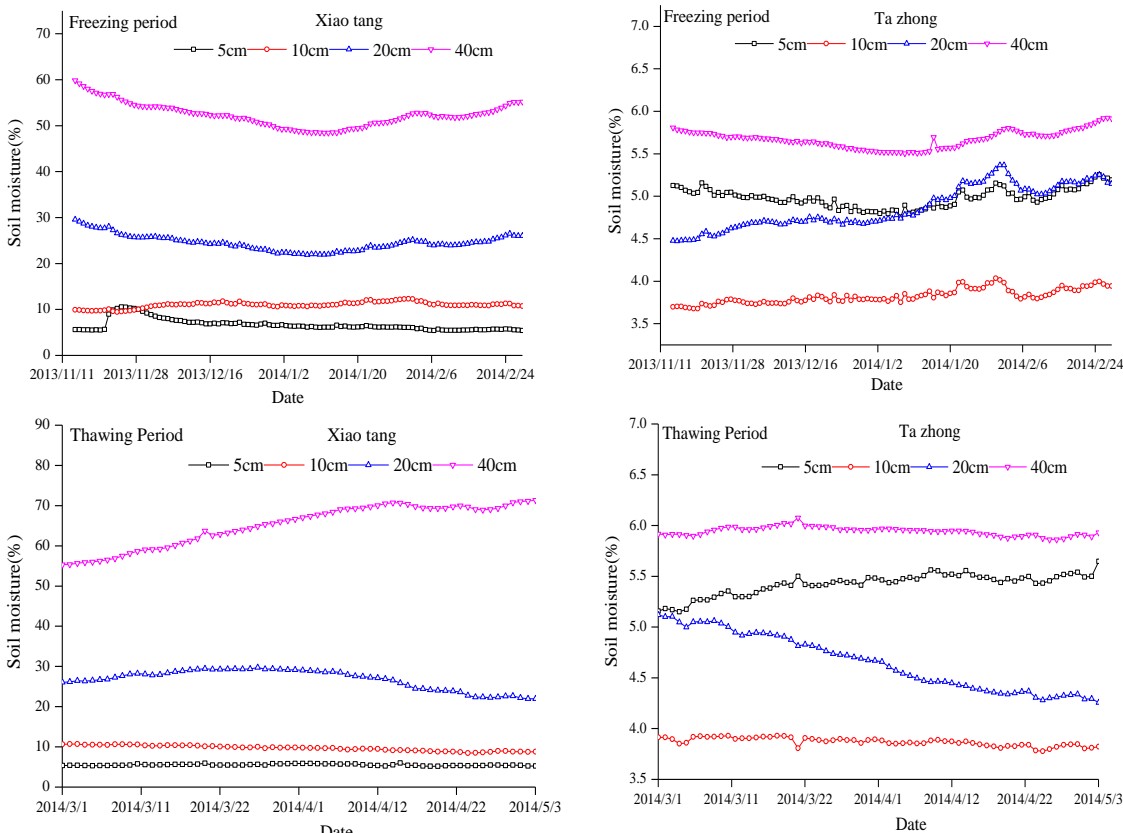

**Figure 4.** Variation of Soil Moisture Content during Freezing and thawing.

The soil moisture content during freezing at 5 cm, 10 cm, 20 cm, and 40 cm depths at the Xiaotang station was 6%, 10%, 29%, and 59%, respectively, and those at the Tazhong station were 5%, 3.6%, 4.4%, and 5.8%), respectively. As the ambient temperature continued to decrease, the freezing front moved downward and the soil water contents at all depths decreased.

In the process of soil freezing and thawing in the desert, soil moisture content has a certain relationship with soil freezing temperature and freezing times. When the physical and chemical properties of the soil remain unchanged and the freezing temperature decreases, the amount of freezing can increase and cause significant changes in soil moisture content. For roads built in the freezing-thawing area, the freezing damage is typically related to the migration of the soil moisture during the freezing-thawing process.

When water from the ambient environment unceasingly seeps into the soil, it can cause the soil moisture content and the permeability to increase, which would in turn reduce its cohesive force. The F-T cycle of the soil would weaken the soil bearing strength and make it deform easily. Consequently, the buildings and roads in the affected area would sink and cause damage. Whilst many scholars have conducted a large number of experiments, the findings are limited. A plausible explanation is that the soil moisture content in the freezing and thawing period can influence the shear strength of the soil foundation [15–19].

As shown in Figure 5, the surface and interior of the roadbed soil have experienced temperature fluctuations because of the frequent usage of the road and the influence of environmental factors (e.g., solar radiation). Due to the different locations, soil temperature on the surface of the roadbed is greatly affected by the ambient temperature, and heat dissipation is easier. The surface temperature rise is much smaller than that of the interior, resulting in a large temperature difference between the internal and external layers, which may eventually lead to overall frost heave. When the soil temperatures at different depths are significantly different from the ambient temperature, cracks can occur from the surface

to the interior, which is consistent with the actual damage. At the same time, slight freezing and thawing can happen when the temperature fluctuates around 0 °C. The driving strength of a vehicle is affected by the amount of freezing and thawing. This can also accelerate the formation of cracks. If the frequency of freezing and thawing is too high, the roadbed soil would be subjected to cracking. If there is liquid water in the subgrade soil, it would cause local frost heave cracking due to the freezing of low temperature water.

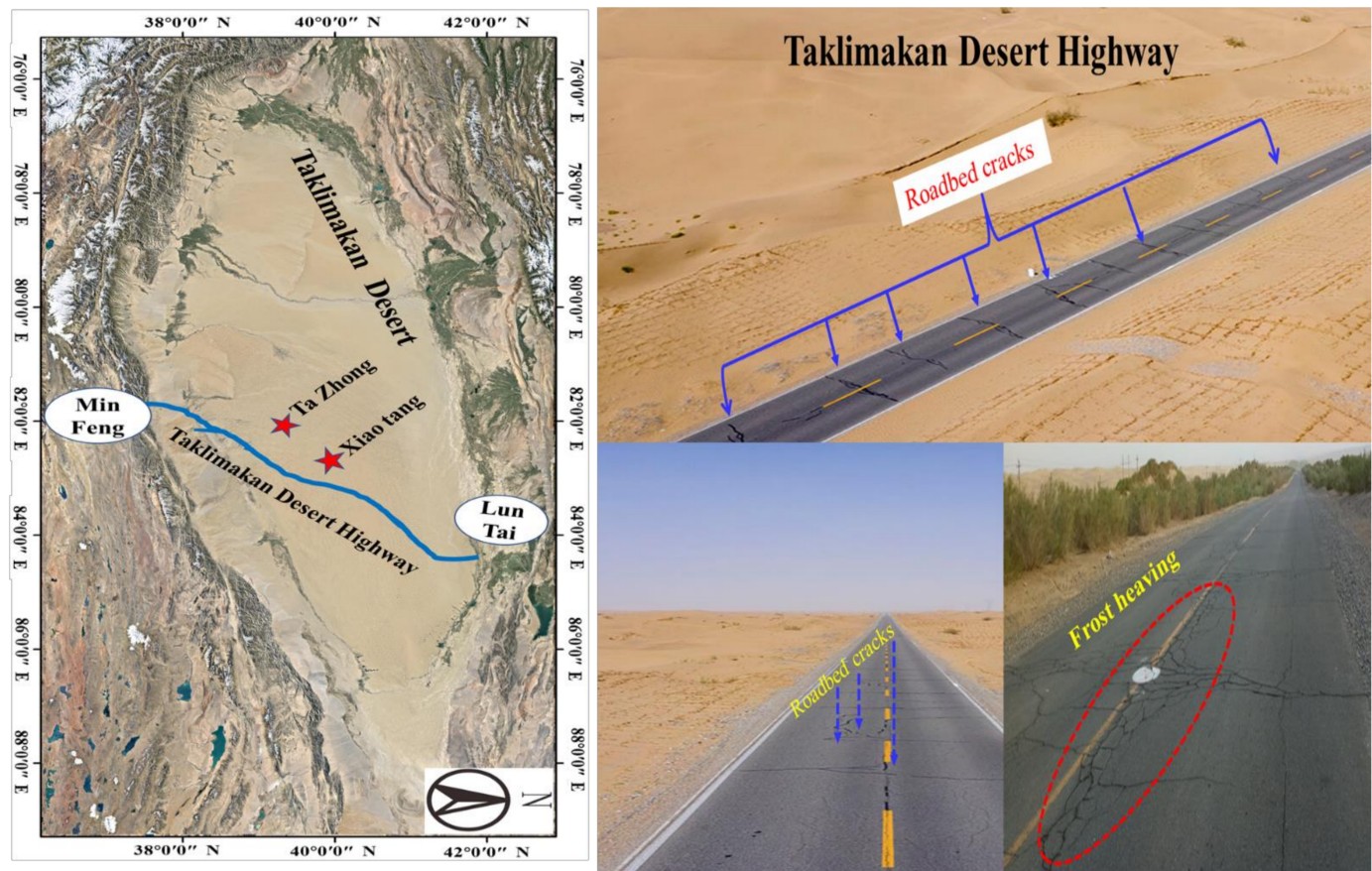

**Figure 5.** Freeze-thaw changes in the soil damage the roadbed.

### 3.3. Soil Water-Heat Coupling Model

The measured soil moisture content ($\theta v$) and temperature ($T_S$) data were used to establish a regression model. These two parameters were connected to establish a coupling model of soil moisture content and temperature at soil depths of 20 and 40 cm, respectively. Here we compare the observed values and analyze the error of the model. During the freezing-thawing cycle of the soil, the regression model constructed using soil moisture content and temperature can be expressed by a unified equation (see Table 1). The fitted soil moisture content and soil temperature are displayed in Figure 6.

**Table 1.** Regression equation fitting of soil moisture content and soil temperature.

| Soil Depth/cm | Soil Types | Regression Equation | Residual Sum of Squares | R² |
|---|---|---|---|---|
| 20 cm | Seasonally frozen soil | $\theta_v = 25.04e^{0.0115Ts}$ | 0.47 | 0.927 |
| 40 cm | | $\theta_v = 50.91e^{0.0085Ts}$ | 0.18 | 0.943 |

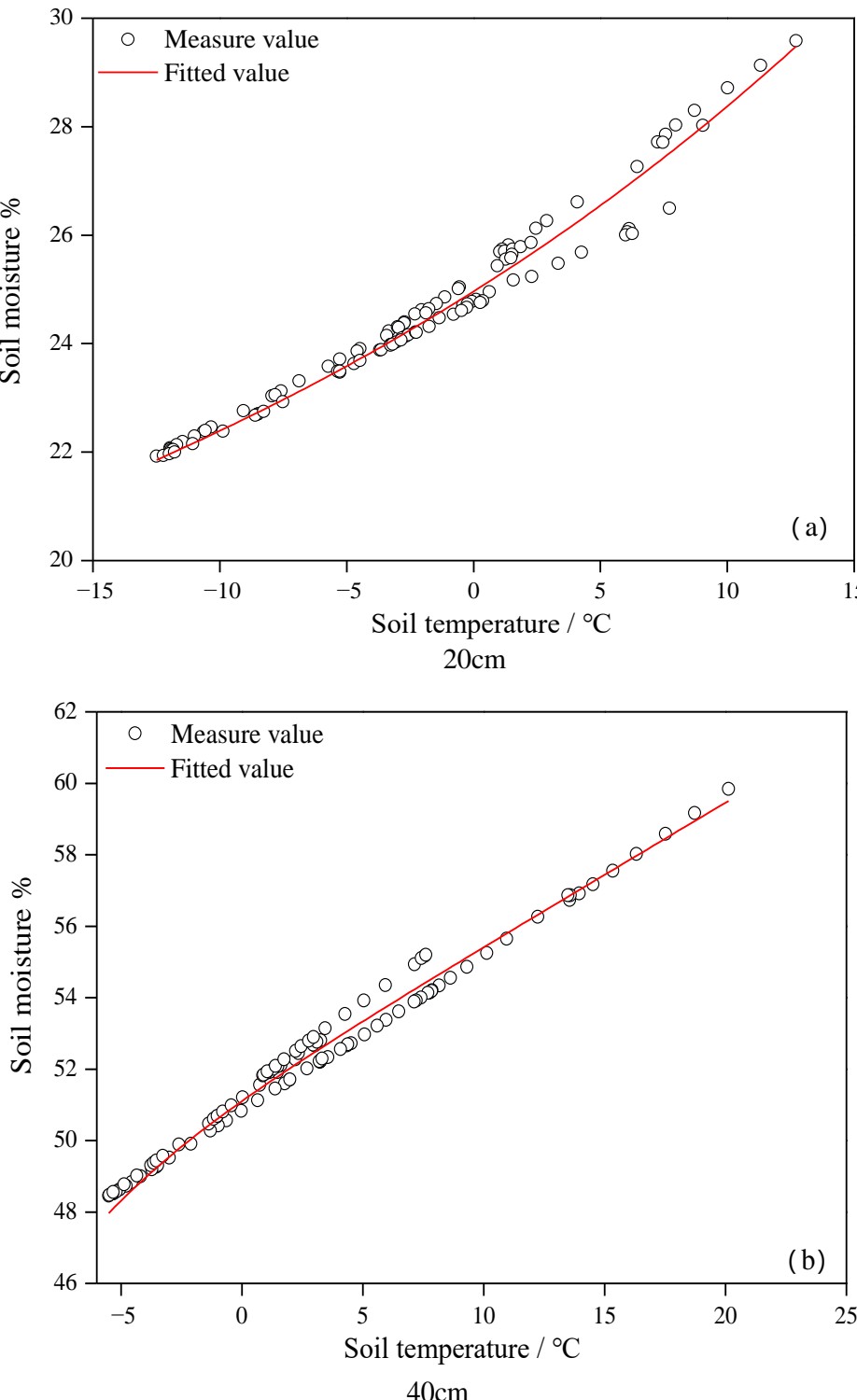

**Figure 6.** Curve of coupling relationship between soil temperature and moisture content.

The models of soil moisture content and soil temperature are presented in Table 1. These regression models are characterized by exponential functions, and the $R^2$ at 20 cm and 40 cm are 0.927 and 0.943, respectively. In statistical analysis, $R^2$ greater than 0.8 is generally considered to be highly correlated. Among them, the correlation between soil moisture content and temperature is high, at 20 cm depth, and the coupling and interaction between the two are strong. The shallow soil at 20 cm depth is covered by snow. Because the ambient temperature, ambient humidity, atmospheric radiation, and other ambient

influences are greater, the hydrothermal coupling relationship of the surface soil is relatively weak. Regarding the soils at 5 cm and 10 cm depths, the correlation between water content and temperature is not high, probably due to their closeness to the surface, combined with the surface radiation flux and the influence of diurnal temperature difference.

### 3.4. The Variation of Soil Flux by F-T

Figure 7 shows that the soils at the Tazhong and Xiaotang stations have experienced four processes: freezing in autumn, cooling in winter, warming in spring, and melting in summer. $R_n$, H, and LE responded differently to changes in ambient temperature. The initial freezing times of these stations were similar, but the Rn of the Xiaotang station was higher than that of the Tazhong station in both freezing and thawing phases, which was about 415 W/m$^2$. This is probably because the Xiaotang station is close to the Tarim River and *Populus euphratica* forest, where the soil temperature is directly affected by solar radiation, and different surface cover can directly influence the amount of solar radiation absorbed by the soil. For H, the Xiaotang and Tazhong stations showed a similar pattern, i.e., the value bottomed (102 W/m$^2$) when completely melted and peaked (295 W/m$^2$) when frozen, and the rest was basically around 180 W/m$^2$. The LE at the Tazhong station was slightly higher than the Xiaotang station, but it reversed during the initial freezing period and after complete freezing. This is probably because, after complete freezing, the internal heat was reduced and the temperature was increased. This heat preservation effect is quite consistent with previous simulation results [10]. During the complete thawing period, the soil moisture in the shallow layer of the surface began to increase (Figure 7), and the evapotranspiration of the surface soil to the atmosphere also increased. The variation of LE was similar to that of $R_n$, which explains the dominance of LE in the surface energy distribution. Our results show that the soil water content is transferred to the atmosphere through solar radiation and evaporation, forming water vapor and heat energy. Therefore, the surface soil heat flux is reduced, which inhibits the increase of soil temperature, resulting in a decrease of land-atmosphere temperature.

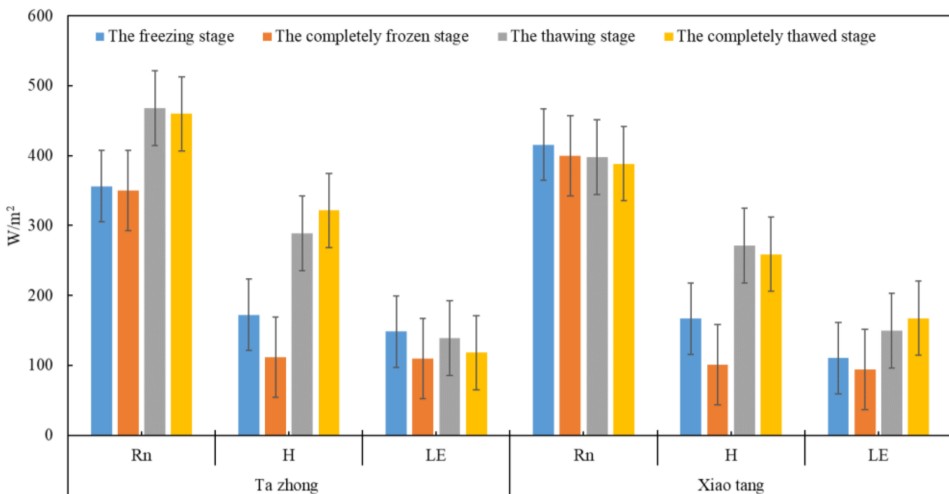

**Figure 7.** Flux variation at field observation stations during the period November 2013 to May 2014.

By studying the long-term field observations in the Taklimakan Desert, we find that snow covers the surface in winter, which changes the solar radiation, thereby affecting sensible and latent heat fluxes. In addition, the diurnal variation of $R_n$'s can be significant due to the large change in weather environment. During the F-T stage, the Rn's flux at the Xiaotang station is higher than that at the Tazhong station.

The moisture content and evaporation of shallow soil in the complete freezing period are generally low, hence a lower LE. The dominance of H in the energy distribution near the ground leads to the consistent variations of H and $R_n$. During the freezing stage, soil

moisture begins to migrate and the moisture content gradually decreases, resulting in the gradual increase of H and the decrease of LE. If LE starts to decrease before soil freezing, it means that soil freezing also affects its flux change, and H begins to increase with Rn before soil thawing. When the soil begins to melt, $R_n$ still increases with the seasons, causing the gradual increase of H and the enhancement of LE. The increase of soil moisture content after thawing continuously contributes to the latent heat flux, and this process continues with the increase of soil thawing time [20]. In addition, the water migration in the whole process of soil freezing and thawing always occurs in a continuous cycle. When the soil enters a period of complete freezing, the formed water-resisting layer prevents the downward infiltration of external water. In this sense, water is mainly distributed on the surface of the soil, and it is more likely for evaporation to occur. In short, the soil F-T cycle in the desert area has a certain impact on the changes of surface heat fluxes.

When the soil is completely frozen, the soil temperature at 5 cm depth is obviously below 0 °C. This is due to the incomplete frozen liquid water in the soil at this stage. At this stage, the diurnal variation of LE remains visible despite the relatively small diurnal variation of soil moisture. (Figure 2). Due to the influence of solar radiation, the diurnal variation of H is greater than that of LE. During the period of complete freezing, the diurnal variation of Rn is quite obvious in the change of its flux. This phenomenon might be due to the heat preservation effect of the ice-bearing layer formed during the complete freezing of the soil and the surface albedo during the snowfall.

During the soil freezing and thawing period, the soil moisture content at 5 cm depth at the Tazhong station was significantly larger than that at the Xiaotang station, and was also greater than the values at 10 cm and 20 cm depths (see Figure 3).

During daytime, the effect of F-T cycles on shallow soil still results in small LE. Moreover, soil moisture content decreases first and then increases during the day (contrary to the diurnal variation of net radiation). In other words, when the net radiation is high, the water content in the surface soil is low, and these negative correlation changes inhibit the increase of LE in the daytime [21]. The results show that, during the soil freezing period, the change of LE was significant. A plausible explanation for this difference is that the soil moisture content in the model simulation was different from the field measured data. The diurnal variation of H was much greater in the early thawing period compared to the complete thawing period. The reason for this is that, when completely thawed, the soil moisture content would increase, and solar radiation is mainly distributed to the evaporation latent heat.

## 4. Discussion and Conclusions

During the freezing period, as the amount of snow cover increases and the ambient temperature decreases, the liquid water in the soil will gradually turn into solid ice and the shallow and middle soil layers will freeze. At the same time, as the degree of freezing increases, the soil freezing front gradually moves down and the soil temperature and liquid water content trend downward. During the thawing period, as the ambient temperature rises, with the transformation from freezing period to complete thawing period, the temperature and soil moisture content trend upward. During the freezing and thawing process, the variations of soil temperature and moisture content trend downward with the increase in soil depth (0–40 cm). The temperature at each depth of the soil layer decreases and varies with the ambient temperature. The temperature change trend remains the same, but the middle and deep soil temperatures have certain hysteresis with the environmental temperature changes. The moisture content of deep soil remains basically unchanged during the freezing and thawing process, and it hardly changes with the fluctuations of the ambient temperature. The coupling relationship between soil moisture content and temperature demonstrated a strong exponential function at 20 cm and 40 cm depths. By comparing the coupling relationship model between 20 cm and 40 cm soil moisture and temperature, we found that, as the depth of the soil layer increases, the effect of the ambient environment on soil moisture and the temperature gradually weakens and the coupling

effect of the two will be enhanced. Moreover, there is no correlation between the moisture content of the middle and deep soil and the temperature.

In this study, our field measurements only lasted for two years. Further research would be needed on the accumulation of water at the freezing-thawing interface of the roadbed of the desert highway. In particular, regarding the freezing and thawing hazards of desert highways, sensors should be installed on both sides of the roadbed. In this way, it would be possible to gather some long-term soil moisture migration data for supporting engineering design.

**Author Contributions:** Conceived and designed the experiments: L.X. and L.H. Performed the experiments: K.Y. Software: C.H. All authors have read and agreed to the published version of the manuscript.

**Funding:** This work was supported by "Tianshan Youth Talents (Xinjiang)" Plan Project (2019Q037), the Second Tibetan Plateau Scientific Expedition and Research (STEP) program (grant no.2019QZKK010206) and General scientific research project of Xinjiang Meteorological Bureau (MS201907).

**Acknowledgments:** This work was supported by "Tianshan Youth Talents (Xinjiang)" Plan Project (2019Q037), the Second Tibetan Plateau Scientific Expedition and Research (STEP) program (grant no.2019QZKK010206) and General scientific research project of Xinjiang Meteorological Bureau (MS201907).

**Conflicts of Interest:** The authors declare no conflict of interest.

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
