# Peer review of "Hydrothermal Effects of Freeze-Thaw in the Taklimakan Desert"

_sustainability, doi:10.3390/su13031292_

Round 1

Reviewer 1 Report

Dear Authors,

The work is interesting. Yet, there exists enough room for improvement.
Please highlight the work's novelty in a better way. Besides, conduct a thorough literature review in order to point out what your study brings to the current body of knowledge.

Given the fact that you're talking about a challenge "freezing soil", it would be great if you could come with suggestions to tackle such a challenge.

The English language level requires some improvement. Please have the manuscript proofread by a ntaive English speaker.

Another remark, the incorporation of equations seems distorting. Please improve this point.

Best Regards,

Author Response

Reviewer 1

Comments and Suggestions for Authors

Dear Authors

Comment 1:

The work is interesting. Yet, there exists enough room for improvement.

Please highlight the work's novelty in a better way. Besides, conduct a thorough literature review in order to point out what your study brings to the current body of knowledge.

Reply 1: Thanks for your suggestion. The aim of the work is to studying the relationship between soil moisture content and temperature from 0 to 40cm, the basic parameters obtained can provide support for designing the thickness of subgrade cushion and paving. Through the analysis of field observation data, this paper expounds the mechanism of soil freezing and thawing in desert areas, and provides technical support for subgrade design and even post-repair engineering. It is not only a desert highway in Xinjiang, China, but also a country with similar desert highways along the Belt and Road.

Comment 2:

Given the fact that you're talking about a challenge "freezing soil", it would be great if you could come with suggestions to tackle such a challenge.

Reply 2: Thanks for your suggestion. In the early stage, the state-level observation stations in the Taklimakan Desert are Tazhong and Xiaotang. The previous studies mainly focus on the observation of desert wind and sand, dust storms, aerosols and other aspects. With the change of global climate, soil freezing and thawing in desert areas has also occurred frequently. Since the completion of desert highway in 2005, pavement cracks have often occurred along with the freezing and thawing cycle of subgrade, which seriously affects the transportation of oil and the north and south of Xinjiang. Therefore, at present, we are beginning to pay attention to this research, strive to obtain a large number of experimental data, and propose solutions for ‘freezing soil’.

Comment 3:

The English language level requires some improvement. Please have the manuscript proofread by a ntaive English speaker.

Reply 3: Thanks for your suggestion. The manuscript has been coloured by native English speakers.

Comment 4:

Another remark, the incorporation of equations seems distorting. Please improve this point.

Reply 4: Thanks for your suggestion. The equation has been re-verified and modified.

Reviewer 2 Report

Overall, the content is present. Flow of logic makes sense, but the text is full of typing errors, grammar errors, and even a few misspelled words. Starting at line 22 the sentences start to become incomplete. There are a lot of grammar issues with capitalization, and lack of. Sentences start and end inconsistently, and several periods end a sentence with no space afterwards. 

Figure 1 needs further explanation. It is difficult to understand terrain with the non-continuous multi-color ramp. No reference of the probes with the figure or directly in text. They should be in a separate image or clearly explained and referenced. Beyond fig 1, some figure references in bold, yet others not; inconsistent. 

Figure 7, recommend the legend text be more concise. Do not need to be wrote in sentence structure

In methods, formatting issues with the variables. Extra spacing with some and inconsistent use of other variables like at line 140 with Ground heat flux (Rn), but then Rn not used in text, but rather G of zero.

After line 153, even more spacing issues and inconsistencies. 

Seems to be a jump from the idea of solar energy influence to effect on soil temperatures, freezing and upheaval. Latent heat flux is mentioned in methods, but then not plotted or discussed. Then solar energy units presumed to be LE show up in figure 7 with explanation to how those data were produced.

Unclear from the text the issue being solved or how the research adds to current knowledge. Quick Google search reveals that to prevent or slow these types of road failures, the roads need to be built so the water volume level is reduced or even eliminated. The solution to protecting roads appears to be better drainage design to keep the water out and away so freezing upheaval is reduced, even eliminated.

Author Response

Reviewer 2

Comments and Suggestions for Authors

Comment 1:

Overall, the content is present. Flow of logic makes sense, but the text is full of typing errors, grammar errors, and even a few misspelled words. Starting at line 22 the sentences start to become incomplete. There are a lot of grammar issues with capitalization, and lack of. Sentences start and end inconsistently, and several periods end a sentence with no space afterwards.

Reply 1: Thanks for your suggestion. The manuscript has been coloured by native English speakers.

Comment 2:

Figure 1 needs further explanation. It is difficult to understand terrain with the non-continuous multi-color ramp. No reference of the probes with the figure or directly in text. They should be in a separate image or clearly explained and referenced. Beyond fig 1, some figure references in bold, yet others not; inconsistent.

Reply 2: Thanks for your suggestion. The map of the study area has been revised, namely, Fig. 1. Details such as digital bold have been fully verified and modified..

Comment 3:

Figure 7, recommend the legend text be more concise. Do not need to be wrote in sentence structure

In methods, formatting issues with the variables. Extra spacing with some and inconsistent use of other variables like at line 140 with Ground heat flux (Rn), but then Rn not used in text, but rather G of zero.

Reply 3: Thanks for your suggestion. Figure 7 has been modified, variables and spacing have been modified.

Comment 4:

After line 153, even more spacing issues and inconsistencies.

Reply 4: Thanks for your suggestion. After line 153, the more spacing issues have been modified

Comment 5:

Seems to be a jump from the idea of solar energy influence to effect on soil temperatures, freezing and upheaval. Latent heat flux is mentioned in methods, but then not plotted or discussed. Then solar energy units presumed to be LE show up in figure 7 with explanation to how those data were produced.

Reply 5: Thanks for your suggestion. The missing part of the content has been added to the manuscript.

Comment 6:

Unclear from the text the issue being solved or how the research adds to current knowledge. Quick Google search reveals that to prevent or slow these types of road failures, the roads need to be built so the water volume level is reduced or even eliminated. The solution to protecting roads appears to be better drainage design to keep the water out and away so freezing upheaval is reduced, even eliminated.

Reply 6: Thanks for your suggestion. Studies have shown that water migration along with the freezing and thawing cycle of subgrade surface will cause frost heave and thaw settlement diseases, Precipitation will also affect the subgrade hydrothermal condition. At present, there are many studies on the temperature field of subgrade, and there are few studies on the distribution of water migration on the surface of subgrade in the field and its influence on the temperature field of subgrade. The basic data obtained in this study will provide basic data and support for road rehabilitation and other repairs

Reviewer 3 Report

In this manuscript, Lu et al reported the freezing-thawing process in the Taklimakan Desert through observation data based on Tazhong and Xiaotang sites from November 11, 2013 to May 2014. Although it is important to demonstrate the effects of freeze-thaw process in desert area, this paper do not provide any insight in this key problem. While the title focuses on the “Hydrothermal mechanism”, but the corresponding discussion is rare, and only the presentation of the observation data. Therefore, I cannot recommend it to be published in Sustainability. Below are my comments:

  1. Why chose Tazhong and Xiaotang sites as studying targets, are they representative in studying the freezing-thawing processes?
  2. What is the main contribution of this study? Is it inspiring to other similar researches? While the authors want to demonstrate the hydrothermal mechanism, what is it?
  3. In Figure 6, what is the relationship between soil moisture and temperature? Linear? But it clearly showed two linearities. While the authors said there was good relationship for a soil depth of 20 and 40 cm, but in the conclusion section, they said “The coupling effect between soil moisture and temperature is mainly manifested in the coupling relationship between water and heat in the shallow soil depth of 5 cm and 20 cm”. It is confusing.
  4. The writing should be improved, a lot of typos were found.

Author Response

Reviewer 3

Comments and Suggestions for Authors

In this manuscript, Lu et al reported the freezing-thawing process in the Taklimakan Desert through observation data based on Tazhong and Xiaotang sites from November 11, 2013 to May 2014. Although it is important to demonstrate the effects of freeze-thaw process in desert area, this paper do not provide any insight in this key problem. While the title focuses on the “Hydrothermal mechanism”, but the corresponding discussion is rare, and only the presentation of the observation data. Therefore, I cannot recommend it to be published in Sustainability. Below are my comments:

Comment 1:

Why chose Tazhong and Xiaotang sites as studying targets, are they representative in studying the freezing-thawing processes?

Reply 1: Thanks for your suggestion. At present, the large-scale observation stations in the Taklimakan Desert mainly include Tazhong Station and Xiaotang Station, which can basically cover the field observation range of the Taklimakan Desert, and the distance between the two stations is about 100 km. Moreover, the above two observation stations are the two observation stations closest to the desert highway, and the data are relatively complete, so they are selected.

Comment 2:

What is the main contribution of this study? Is it inspiring to other similar researches? While the authors want to demonstrate the hydrothermal mechanism, what is it?

Reply 2: Thanks for your suggestion. Since the desert highway was built in 2005, the road transport task is heavy. Due to the freezing and thawing cycle of soil, the roadbed is often damaged, which seriously affects the transportation efficiency of northern and southern Xinjiang. Through the analysis of field observation data, this paper expounds the mechanism of soil freezing and thawing in desert areas, and provides technical support for subgrade design and even post-repair engineering. It is not only a desert highway in Xinjiang, China, but also a country with similar desert highways along the Belt and Road.

In this paper, by studying the relationship between soil moisture content and temperature from 0 to 40cm, the basic parameters obtained can provide support for designing the thickness of subgrade cushion and paving.

Comment 3:

In Figure 6, what is the relationship between soil moisture and temperature? Linear? But it clearly showed two linearities. While the authors said there was good relationship for a soil depth of 20 and 40 cm, but in the conclusion section, they said “The coupling effect between soil moisture and temperature is mainly manifested in the coupling relationship between water and heat in the shallow soil depth of 5 cm and 20 cm”. It is confusing.

Reply 3: Thanks for your suggestion. The relationship between soil moisture and temperature is an exponential function, which should be 20cm and 40cm. Any doubts have been revised in the text.

Comment 4:

The writing should be improved, a lot of typos were found.

Reply 4: Thanks for your suggestion. The manuscript has been coloured by native English speakers.

Round 2

Reviewer 1 Report

Dear Editor,

Dear Authors,

The manuscript is revised as suggested.

Best Regards,

Reviewer 3 Report

The authors addressed all my concerns, and the quality of manuscript was improved a lot after revision. It can be accepted in present version.